# Effect of Speed Threshold Approaches for Evaluation of External Load in Male Basketball Players

**DOI:** 10.3390/s25196085

**Published:** 2025-10-02

**Authors:** Abel Ruiz-Álvarez, Anthony S. Leicht, Alejandro Vaquera, Miguel-Ángel Gómez-Ruano

**Affiliations:** 1Faculty of Physical Activity and Sport Sciences, Universidad Politécnica de Madrid, 28040 Madrid, Spain; abel.ruiz.alvarez@alumnos.upm.es; 2Sport and Exercise Science, James Cook University, Townsville 4810, Australia; anthony.leicht@jcu.edu.au; 3Faculty of Physical Activity and Sport Sciences, Universidad de León, 24075 León, Spain; avaqj@unileon.es

**Keywords:** team sports, technology, players, micro-sensors

## Abstract

Arbitrary zones are commonly used to describe and monitor external load (EL) during training and competitions. However, in recent years, relative speed zones have gained interest as they allow a more detailed description of the demands of each individual player, with their benefits largely unknown. This study aimed to (i) identify differences in EL methodological approaches using arbitrary and relative running speed zones; (ii) examine the effect of the methodological approaches to identify fast and slow basketball players during competition and training; and (iii) determine the effect of the season stage on the methodological approaches. Twelve players from a Spanish fourth-division basketball team were observed for a full season of matches and training using inertial devices with ultra-wideband indoor tracking technology and micro-sensors. Relative velocity zones were based on the maximum velocity achieved during each match quarter and were retrospectively recalculated into four zones. A linear mixed model (LMM) compared fast and slow players based on speed profiles between arbitrary and relative thresholds and during each competition stage. All players surpassed peak speeds of 24 km·h^−1^ during the season, exceeding typical values reported in elite basketball (20–24.5 km·h^−1^). Arbitrary thresholds produced greater distances in high-speed running (Zones 3 and 4) and yielded lower values in low-speed activity (Zone 1), with differences of ~100 m and ~120–250 m, respectively (*p* < 0.001), particularly for fast-profile players. These discrepancies were consistent across most stages of the season, although relative zones better captured variations in Zone 1 across time. Training sessions also elicited +8.7% to +40.7% greater distances > 18 km·h^−1^ compared to matches. The speed zone methodology substantially influenced EL estimates and affected how player EL was interpreted across time. Arbitrary and relative approaches offer unique applications, with coaches and sport scientists encouraged to be aware that using a one-size-fits-all approach may lead to misrepresentation of individual player demands, especially when tracking changes in performance or managing fatigue throughout a competitive season.

## 1. Introduction

There is a consensus among sport scientists on the importance of load monitoring in the context of athletic training, as it is the most important variable that is manipulated to obtain the desired training response [1,2]. The organization, quality, and quantity of exercise (training plan) determine the external load (EL), which is defined as the physical work prescribed in the training plan [1,3]. Tracking systems’ data has endless applications; however, these can be summarized within three main functions: (a) describing, (b) planning, and (c) monitoring competitive and training loads to support decision-making related to player performance and availability [4,5]. To implement any technology to monitor EL, sport scientists must consider and understand several key aspects that will affect the quality of the results, especially any decision-making system [6]. Some of these key aspects are related to the variables used, the training intensity zones and thresholds, the criteria for defining an effort, and the filtering and processing of the data that will determine the results [7,8]. In this context, training intensity is commonly defined as the physical work performed per unit of time [9] and is fundamental to determining the intensity thresholds for determination of physical demands [10,11,12]. Among other reasons, the correct description of intensity during training and matches is essential to describe the prescribed training drills in relation to competitive demands and to optimize periodization for players [4,13]. Although the sensors employed in this study were commercially available, the focus of this work lies in their rigorous methodological application and validation in real-world exercise contexts. This aligns with the scope of the Special Issue, which emphasizes the applied use of sensor technologies to optimize performance and monitoring in sport settings.

Typically, basketball player activity is quantified using arbitrary thresholds with inconsistencies in intensity zones and thresholds observed among common monitoring technologies (i.e., video-based time motion analysis—TMA, microsensors, and local positioning systems—LPS), especially for variables such as velocity, acceleration, and deceleration [8]. Prior studies using LPS have reported variations in arbitrary running speed zones, including the number of zones and the speed limits of each zone [8]. For example, many basketball [14,15,16] studies have combined standing and walking activity within the same zone; however, others classify these as separate zones [8]. Further, several have engaged different intensity thresholds for standing and walking activities, with the initial thresholds ranging from < 2.14 km·h^−1^ to < 7 km·h^−1^ across studies. For higher-intensity zones, most studies have defined “high-speed running” or “high-intensity running” with consistent thresholds of at least ~18 km·h^−1^ [14,15,16,17] or with an upper limit of 24 km·h^−1^ [18,19,20,21,22,23,24]. Similarly, for sprinting activity, most basketball studies have employed a threshold of >24.1 km·h^−1^ [18,19,20,21,22,23,24], with some adopting a threshold of >21 km·h^−1^ [8,25].

Recent basketball studies have used different approaches to establish relative intensity zones [12,26] with various terminology applied, generating different perspectives and conflicts over the semantics of this concept. “Speed profile” [27], “Relative intensity zones” [8], “workload thresholds”, “workload ranges”, “position-specific intensity thresholds” [28], and “Individualized thresholds” [12,29] are the terms most commonly used by researchers to refer to relative speed zones that determine the external demands experienced by players. The selection of appropriate individualization methods based on fitness levels represents a major challenge [30]. In this regard, Clemente et al. [26] emphasized the importance of improving methodological rigor and replicability when applying individualized speed thresholds, recommending that studies consider population-specific and contextual factors rather than simply proposing alternatives to arbitrary cut-offs. The establishment of relative intensity zones for speed, acceleration, deceleration, player load and impact traversed in basketball has been studied previously. In these studies, different statistical techniques such as k-means and/or two-step clustering [11,12,27,28] or standard deviations [31] were used to determine reference values and divide variables into statistically different groups. These methods individualized training zones based on each player or playing position, helping to understand the specific demands during training and matches relative to their physical capacity. They also complement arbitrary thresholds by providing more detailed insights into how intensely a player trains or competes in relation to their own physical limits [26]. However, examination of arbitrary and relative thresholds for speed zones has been rarely investigated in the same study of basketball players.

Current research on individualizing relative speed zones in basketball has faced several challenges. For example, while most studies focused on specific contexts such as official matches [27], youth tournament matches [31], pre-season elite tournament [12,28] or training matches [11], there was limited integration of data beyond these specific contexts with examination of a full regular season and training sessions likely to incorporate higher-intensity running demands than single matches [32]. Additionally, efforts to individualize speed zones by playing positions [12] have been complicated, with players able to shift between positions within and between matches due to injuries, tactical adjustments, or other factors [33,34]. To address these gaps, it is essential to develop speed zones that reflect each player’s physical condition, independent of their position. Notably, García et al. [32] highlighted the relevance of high-speed running (>18 km·h^−1^) as a differentiating variable, with this being the only one that surpassed match demands during specific training sessions. Another critical consideration is whether relative speed zones should be updated throughout the season to account for fitness fluctuations (i.e., improvements with training or decrements due to injury or lack of match-play).

Consequently, this study aimed to (i) identify differences in EL methodological approaches using arbitrary and relative running speed zones; (ii) examine the effect of the methodological approaches to identify fast and slow basketball players during competition and training; and (iii) determine the effect of the season stage on the methodological approaches. Importantly, this work will integrate longitudinal data from both training and competition, offering a broader perspective on how threshold selection influences EL quantification and providing more precise, adaptable performance metrics to support the physical demands of basketball players.

## 2. Materials and Methods

### 2.1. Design

A non-experimental observational design was employed for this study [35]. Recordings of EL data from basketball players during 13 matches and 45 basketball training sessions within the 2021–2022 competitive season (Spanish Amateur Basketball League—LIGA EBA) were collected. The independent variables were the activity (match or training), the stages of the season (Stage 1 = first 12 weeks, Stage 2 = next 8 weeks, and Stage 3 = last 8 weeks), and the player’s speed profile (Slowest And Fastest, which was established by k-means cluster analysis to reveal natural groupings as follows: Slowest, four Centers for a cluster center of 25.3 km·h^−1^; Fastest, three Guards, four Forwards and one Center for a cluster center of 28.8 km·h^−1^). The average peak speed achieved in each of the three season stages was used for cluster analysis. The dependent variables were the EL variables (i.e., total distance in each speed zone) monitored by LPS technology. All training sessions and matches occurred on the same hardwood basketball court under similar internal conditions.

### 2.2. Participants

Male basketball players (n = 12, age: 21.3 ± 5.3, height: 196.5 ± 6.8 cm, body mass: 88.5 ± 10.8 kg) from the same team competing in the fourth division of the Spanish Amateur Basketball League (LIGA EBA) were monitored throughout the 2021–2022 competitive season (from October to April) [36,37]. For inclusion in the study, players (Guards n = 3, Forwards n = 4, and Centers n = 5) competed in at least one match per week (i.e., >29% of the official match time). Players completed at least fifteen minutes of live court time per match or training session. In addition, data collected during the pre-match warm-up, rest periods between quarters and time-outs were not considered. All participants were informed of the purpose, risks and benefits of the study in accordance with the Declaration of Helsinki [38] and provided written informed consent as approved by the local institutional human research ethics body.

An a priori sample size calculation was conducted to determine the participant number for the applied Linear Mixed Model (LMM) design. Parameters were set using G*Power software (version 3.1.9.7), considering a repeated-measures analysis with both fixed (activity type, season stage, and speed profile) and random (player identity) effects. The calculation assumed a medium effect size (f = 0.25), an alpha level of 0.05, and statistical power of 0.80. These criteria were based on previous research involving external load comparisons in team sports using similar statistical frameworks [39,40]. The model included two between-group categories (fast and slow speed profiles), repeated measurements across four speed zones, and multiple observations per player across training and match conditions during different stages of the season. Under these assumptions, the required minimum sample size was estimated to be 12 participants.

On average, participants trained four times a week and played an official match every week, as well as strength, conditioning and individual technique sessions. All playing positions were defined by the head coach, with players assigned to one of three positional groups: Guards, Forwards, and Centers [41]. Overall, 565 recordings were included in the analysis: 439 training and 126 match samples amongst the 12 players. For the playing positions, 151 samples were from Guards, 171 from Forwards, and 243 from Centers. For the stages of season, 193 samples were from Stage 1 (October, November, and December), 201 samples from Stage 2 (January and February), and 171 from Stage 3 (March and April).

### 2.3. Variables

The following parameters were examined within each recording: total distance (m) is the meters traveled by the players while on court; distance per zone (m) is the meters traveled in each speed zone (for both relative and arbitrary speed zones); and peak speed is the maximum speed (km·h^−1^) recorded by an player during the live time of the match or training.

The speeds delimiting the arbitrary speed zones and the percentages for the relative bands are specified in Table 1. Recent reviews highlighted that most LPS studies in basketball used a speed threshold of 18 km·h^−1^ to define high-speed running, with sprinting or very high-speed running occasionally classified as speeds above 24.1 km·h^−1^ [8]. However, the dimensions of a basketball court may limit players from fully reaching or sustaining their maximum speed [14]. Consequently, many studies have adopted four speed zones, with Zone 4 defined as >18 km·h^−1^, reasoning that maximum speeds recorded in competition (24.5 ± 1.7 km·h^−1^) show low variability between players and represent a minimal percentage of overall running, making their impact on performance potentially negligible [8,14,42]. In the current study, the peak speed for each participant was identified and used for all zone calculations within training and match records. Additionally, four relative speed zones were established based upon an arbitrary peak speed of 24.1 km·h^−1^ (classified previously as sprint activity) to allow comparisons with prior arbitrary four-zone models [14,15,16,17]. Therefore, the relative zones considered were Zone 1 (<29%), Zone 2 (>29%–58.0%), Zone 3 (>58.1%–75%), and Zone 4 (>75%) with respect to the player’s peak speed [39,40]. Relative speed zones were examined during each of the three stages of the season.

### 2.4. Procedures

All basketball players were fitted with an LPS device (Vector S7; Catapult Sports, Melbourne, Australia), part of a commercially available system that has been extensively validated for indoor team sports environments [39,40]. The units were inserted into a fitted neoprene vest under regular playing attire and positioned on the upper thoracic spine between the scapulae [43].

The Vector S7 device contained microsensor technology consisting of an accelerometer (±16 g, 100 Hz), magnetometer (±4.900 µT, 100 Hz), and gyroscope (up to 2000 deg/s, 100 Hz). Each device sampled at 10 Hz within an ultra-wideband as part of a 4 GHz transmitting system equipped with 24 anchors positioned around the perimeter of the stadium. This LPS technology was reported to be valid in measuring distance [43,44,45,46] speed, accelerations, decelerations [43,45], and PlayerLoad [47]. This device selection aligned with recent recommendations for integrating validated commercial technologies to enhance methodological robustness and ecological validity in sports science research [39,40].

All players were familiar with monitoring technology, as they wore the devices for a minimum of ten days before the start of the study, during the pre-season period. Devices were turned on ~20–40 min before the warm-up phase prior to each game or training. In our protocol, players used the same device throughout the season to control inter-unit variability, and monitoring was consistently conducted under standardized conditions to ensure data accuracy [48].

After the conclusion of each event, arbitrary intensity zone data were extracted using OpenField software (version 3.3.1). For the relative intensity zones, the maximum speed (in km·h^−1^) reached by each player during each stage was recorded, and the relative intensity zones were determined retrospectively using defined thresholds (Table 2). For clarity, definitions of key concepts and monitoring variables are summarized in Appendix A. All LPS data were exported to Microsoft Excel (version 16.0, Microsoft Corporation, Redmond, WA, USA) for further analysis.

### 2.5. Statistical Analysis

All results are reported as mean ± standard deviation (SD). Data normality was assessed using the Shapiro–Wilk test. A repeated-measure LMM was applied, with player identity included as a random effect and speed profile, activity type, and season stage as fixed effects. Model robustness when accounting for random effects was evaluated using both R^2^ marginal and R^2^ conditional, intraclass correlation coefficients (ICC). Results are presented with 95% confidence intervals (CI) and coefficient of variation (CV) where appropriate. Differences were considered statistically significant at a *p* < 0.05. In addition, Cohen’s *d* effect sizes (ES) were calculated for all pairwise comparisons as follows: values < 0.20 indicated no effect; ≥0.21 to 0.49 denoted a small effect; ≥0.50 to 0.79 indicated a moderate effect; and ≥0.80 indicated a large effect [49]. IBM SPSS for Windows (version 23, IBM Corporation, Armonk, NY, USA) was used for clustering identification of the players’ speed profile. The statistical programme Jamovi Project for Windows (version 2.4.11) was used for other analyses.

## 3. Results

Based upon the activity cluster analysis (Table 3), the slow players exhibited significantly greater distance traveled within Zone 4 during the match (Table 3), based upon the arbitrary compared to the relative approach. Similar methodological approach differences were noted for Zone 4 results for the slow players during training (Table 3). Similarly, greater distances traveled within Zones 3 and 4 using the arbitrary approach were observed during matches and training for fast players (Table 3). In contrast, the relative approach resulted in greater distance traveled within Zone 1 during training and matches for fast players (Table 3). The use of relative and arbitrary approaches resulted in similar distances traveled within other speed zones during matches and training (Table 3). In addition, the distribution of variables were presented in Appendix A.

When considering season stages (Table 4), the distance traveled within Zone 4 was significantly greater via the arbitrary compared to the relative approach during Stages 1 and 2, and the entire season for the slow players. Greater distances traveled within Zones 3 and 4 using the arbitrary approach were also observed during all stages and the entire season for fast players (Table 3). In contrast, greater distance traveled within Zone 1 using the relative approach was observed during Stages 1 and 2 and the entire season for fast players (Table 3). The use of relative and arbitrary approaches resulted in similar distances traveled within other speed zones between season stages (Table 4). Additionally, the results of distribution of variables were presented in Appendix A.

## 4. Discussion

This study compared arbitrary and relative speed zone approaches to quantify EL in basketball over a full season. The main findings indicated that arbitrary thresholds systematically overestimated high-speed running (Zones 3 and 4) and underestimated low-speed activity (Zone 1), particularly in fast-profile players and regardless of season stage. These discrepancies were less pronounced for slow-profile players. The results indicated that the choice of speed zone methodology substantially influenced EL estimates and may affect how player workload is interpreted across time. Coaches and sport scientists should be aware that using a one-size-fits-all approach may lead to misrepresentation of individual player demands, especially when tracking changes in performance or managing fatigue throughout a competitive season.

### 4.1. Activity-Based: Training vs. Competition

The current results demonstrated that the use of arbitrary speed zones led to significantly greater distances covered in Zones 3 and 4 during both training and competition (*p* < 0.001). One potential explanation was the high peak speeds observed in our sample, with all players surpassing 24 km·h^−1^ at some point during the season, values that exceed the typical peak velocities (20.0–24.5 km·h^−1^) reported in junior and senior elite basketball populations [11,14,27,28,32,42,50]. This discrepancy may partly account for the overestimation of high-speed efforts when using fixed, arbitrary thresholds, especially for faster players.

These findings must also be considered within the context of training design and coaching practices. Coaches often manipulate variables such as opposition format, space constraints, number of players, or playing rules to target specific technical, tactical, or physical objectives [23,51,52,53]. For example, García et al. (2022) reported that players covered more distance at speeds >18 km·h^−1^ during certain training sessions compared to official matches—particularly on Match Day -4 and Match Day -3, which typically involve more demanding drills [32]. This highlights the importance of considering training context when interpreting EL data, as players may reach higher intensities within training session compared to competitive match play. Moreover, the differences between Match Day -3 and matches ranged from +8.7% to +40.7% for distance run at > 18 km·h^−1^, perhaps because the drills required players to run around the court at high intensities for periods longer than one minute [32]. Therefore, incorporating individualized relative speed zones during training sessions (like that of the current study) may be more beneficial for coaches as it enables relevant design when preparing for matches [54], while managing potential risks like hamstring injury [55]. Therefore, it is important to determine the appropriate training zones (arbitrary or relative) for more accurate description, programming and monitoring of volume and intensity within the periodized microcycle plan [53,56]. Training individualization based upon relative speed zones could be an effective means to optimize players’ preparation and performance for competition [54] while also addressing concerns about “maladaptations” or overtraining [12,26,57].

### 4.2. Stages of the Season

Our results showed that relative speed zones offered a more individualized representation of EL, particularly by better capturing low-speed efforts (Zone 1) in faster players throughout the season. However, differences across the various stages of the season were relatively consistent, with similar trends observed in the comparison between the arbitrary and relative approaches. Although it is often suggested that relative thresholds should be updated to reflect changes in a player’s fitness (e.g., due to injury or performance variation), our data did not provide strong evidence that seasonal changes in fitness significantly influenced the EL distributions when using fixed relative zones. Therefore, while the concept of updating relative thresholds remains theoretically appealing, its practical value requires further investigation.

### 4.3. Strengths and Limitations

The present study was the first to compare arbitrary and relative zones in basketball using both training sessions and matches over a whole season; however, certain limitations must be considered. The results were indicative of the specific competition/training level and may not be representative of teams of different ages, levels of play (e.g., elite vs. sub-elite), and sexes. For example, young players (under 14 years of age), who are influenced by their maturation stage [58], may encounter significant difficulties in reaching speeds above >18 km·h^−1^ and accumulating distance at high speed [50]. The current sample was modest in size and homogeneous, comprising players from a single semi-professional team. Replication of the current study with a larger and more diverse population (e.g., different levels, sexes, and age groups) is recommended to extend the current results. While the sample size of 12 players was determined a priori to detect moderate effect sizes with adequate power, post hoc evaluation revealed that the actual statistical power varied depending on the observed effect magnitude. Specifically, the study achieved high statistical power (≥0.94) for large and very large effects, as observed in key comparisons of EL across speed thresholds. However, statistical power was slightly below the conventional 0.80 threshold for detecting moderate effects (≈ 0.60), suggesting limited sensitivity for more subtle differences. Despite this, the robustness of the results observed for Zones 3 and 4, particularly under the arbitrary approach for fast players, supports the adequacy of the sample size for the primary aims of the study. Future studies aiming to detect smaller effect sizes or evaluate more nuanced interactions may benefit from a larger sample to ensure broader sensitivity. Based on our findings, the use of relative speed zones appears to offer a more individualized and consistent framework for monitoring EL in basketball, particularly when accounting for differences in players’ locomotion profiles. Although arbitrary zones may still be useful in certain comparative contexts (e.g., between teams or levels), relative zones provide more meaningful insight into individual player demands. Therefore, we suggest prioritizing the use of relative zones, particularly in applied settings where training load must be tailored to the physical characteristics and fitness status of each player. Future implementations should also consider how relative thresholds may need to be adjusted for factors such as age, sex, or competitive level to improve the precision of EL monitoring [5,8].

To date, there is no consensus on the optimal method to establish relative thresholds with different approaches and fitness assessment tests used to establish relative speed zones, such as maximal aerobic speed, maximal sprint speed and anaerobic speed reserve, among others [26]. Our study used the same LPS technology to monitor EL during training and competition, which minimized the need to perform fitness tests to modify thresholds and the burden on the player. Subsequently, training and match monitoring via LPS may enable coaches to appropriately update individualized training for optimal adaptations (i.e., relative zones).

### 4.4. Future Lines of Research

Currently, the application of relative training zones for basketball players has received minimal examination, with further work in this and other sports required to assist coaches with training development. Also, given the limited availability of LPS infrastructure for all training and competition basketball courts, future research should continue to investigate alternative metrics such as PlayerLoad [29] accelerations, and decelerations [12,28] from other technology (e.g., inertial measurement unit). The integration of artificial intelligence (AI) and video analysis may also enhance future EL monitoring in basketball. AI can process large volumes of sensor data, identify the intensity and density of high-demand events, and generate more accurate load control models [39,40]. Meanwhile, video provides tactical context by linking each locomotor effort to specific game situations. Together, these technologies offer a more comprehensive and practical approach for coaching staff in training planning and decision-making. Furthermore, for a more complete understanding of performance demands, future studies should consider the inclusion of internal measures of workload, objective and subjective, to EL, and provide a more nuanced perspective of player training. In addition, future research should investigate whether periodic recalibration of relative thresholds (e.g., monthly) enhances sensitivity, whether positional roles influence the differences between methods, and how clustering could be extended to other metrics such as accelerations, decelerations, or PlayerLoad. From an applied perspective, real-time implementation remains challenging, although advances in wearable technologies may soon enable it. For practitioners without access to LPS systems, a practical alternative is to regulate the design of training tasks, particularly limiting the number of full-court drills and activities without opposition [59,60], as these formats tend to elicit greater physiological demands and higher volumes of high-intensity running.

## 5. Conclusions

This study provided novel evidence comparing arbitrary and relative speed zone approaches for evaluating EL in basketball, incorporating both training and match data across an entire competitive season. By classifying players into fast and slow locomotor profiles and analyzing seasonal stages, our findings revealed that arbitrary thresholds overestimated high-speed efforts (Zones 3 and 4), especially for faster players and most season stages. In contrast, relative speed zones yielded greater values within low-intensity efforts (Zone 1), offering a more individualized representation of workload. These differences highlight the potential impact of threshold selection on player monitoring, particularly when interpreting high-speed exposure over time. Given these findings, the use of relative zones may be valuable for tracking individual responses across a season, while arbitrary zones may still provide benchmarking across groups or contexts. Future studies should further investigate how updating relative thresholds over time may refine EL monitoring in team sports.

## Figures and Tables

**Table 1 sensors-25-06085-t001:** Peak velocity recorded during the first, second and third stages of the regular season for each player. Data are presented as mean ± standard deviation (SD).

	1st Stage (n = 17)	2nd Stage (n = 17)	3rd Stage (n = 15)	Player Average
Player 1	28.85 ± 2.09	29.39 ± 3.31	27.46 ± 4.25	28.57 ± 0.81
Player 2	28.94 ± 2.42	28.39 ± 4.27	28.89 ± 4.4	28.76 ± 0.25
Player 3	25.96 ± 1.28	28.62 ± 5.57	24.47 ± 4.41	26.5 ± 1.72
Player 4	28.06 ± 1.92	26.89 ± 3.51	28.23 ± 5.15	27.65 ± 0.6
Player 5	27.8 ± 1.91	27.66 ± 3.95	27.35 ± 4.07	27.6 ± 0.19
Player 6	26.33 ± 2.29	24.39 ± 4.05	25.26 ± 4.52	25.31 ± 0.79
Player 7	25.99 ± 1.23	25.44 ± 4.26	24.71 ± 4.52	25.29 ± 0.52
Player 8	27.99 ± 2.13	28.22 ± 3.91	27.5 ± 3.2	27.96 ± 0.3
Player 9	27.39 ± 0.9	26.65 ± 3.37	30.17 ± 7.82	27.96 ± 1.52
Player 10	26.93 ± 2.05	26.03 ± 3.89	25.19 ± 3.16	26.01 ± 0.71
Player 11	26.78 ± 1.83	29.25 ± 2.83	26.95 ± 4	27.72 ± 1.13
Player 12	28.17 ± 2.34	29.65 ± 4.49	25.9 ± 5.43	28.25 ± 1.54

**Table 2 sensors-25-06085-t002:** Arbitrary and relative speed thresholds for each zone examined in the current study.

Speed Zones	Arbitrary Speed Thresholds (km·h^−1^)	Relative Speed Thresholds (% Player Peak Velocity)
Zone 1 (Standing–Walking)	(<7.0)	<29.0%
Zone 2 (Jogging)	(7.0–14.0)	29.1–58.0%
Zone 3 (Running)	(14.1–18.0)	58.1–75.0%
Zone 4 (High Speed)	(>18.0)	>75.0%

**Table 3 sensors-25-06085-t003:** Arbitrary versus relative demands for players according to their speed group (slow or fast) in training and matches for distance traveled (m). Data are presented as mean ± standard deviation (SD).

Clusters	Event	Speed Zones (m)	Arbitrary (m)	Arbitrary CV (%)	Relative (m)	Relative CV (%)	Mean Difference (m)	95% CI of Mean Difference	ICC	F	*p*	*d*
Slow	Match	Z1	1361.4 ± 895.6	65.8	1468.5 ± 928.4	63.2	107.1	956–1764	0.153	0.33	0.566	0.12
		Z2	868.7 ± 452.2	52.1	835.1 ± 420.3	50.3	33.6	639–1015	0.143	0.14	0.707	0.08
		Z3	274.9 ± 147.5	53.7	243.6 ± 136.8	56.2	31.3	203.3–305.4	0.086	1.11	0.296	0.22
		Z4 *	86.9 ± 51.4	59.1	45.1 ± 37.8	83.8	41.8	55.1–76.5	0.012	18.6	**<0.001**	0.93
	Training	Z1	2334 ± 875	37.5	2458 ± 849.5	34.6	123.5	2236–2550	0.021	1.56	0.212	0.14
		Z2	827 ± 498	60.2	764.5 ± 471.1	61.6	62.9	751–864.4	0.013	1.37	0.243	0.13
		Z3	207 ± 161	77.8	195.4 ± 157.2	80.5	11.5	179.6–222.1	0.005	0.39	0.531	0.07
		Z4 *	118 ± 114	96.6	69 ± 77.6	112.5	49	69.9–115.2	0.043	19.6	**<0.001**	0.5
Fast	Match	Z1 *	1336 ± 605	45.3	1599.2 ± 670.3	41.9	262.67	1335.3–1602	0.042	7.3	**0.008**	0.41
		Z2	1213 ± 529.7	43.7	1205.2 ± 528.2	43.8	7.65	1057–1327	0.086	0.01	0.923	0.01
		Z3 *	400 ± 187.6	46.9	259.5 ± 138.8	53.5	140.87	290–366.2	0.063	31.8	**<0.001**	0.85
		Z4 *	161 ± 84.1	52.2	47.4 ± 36.8	77.6	113.63	83.7–123.7	0.147	145	**<0.001**	1.75
	Training	Z1 *	2136 ± 779	36.5	2384.4 ± 749.6	31.4	248.5	2119–2393	0.053	16	**<0.001**	0.33
		Z2	1026 ± 502	48.9	959 ± 502.9	52.4	66.6	978–1098.2	0.006	2.03	0.155	0.13
		Z3 *	289 ± 181	62.6	209.4 ± 150.5	71.9	79.9	246–288.9	0.018	57.6	**<0.001**	0.48
		Z4 *	162 ± 137	84.6	60 ± 65.4	109	101.8	94.5–125.7	0.033	213	**<0.001**	0.95

Note: F-statistic Linear Mixed Model; Z: Speed Zone; ICC: Intraclass correlation coefficient; CI: confidence interval; *p*: *p*-value; *d*: Cohen’s *d* effect size between arbitrary and relative demands; CV: coefficient of variation. * Statistical differences between relative and arbitrary approaches and shown in bold.

**Table 4 sensors-25-06085-t004:** Arbitrary versus relative demands of players according to their speed group (slow or fast) during three stages of the season during training and matches for distance traveled (m). Data are presented as mean ± standard deviation (SD).

Clusters	Season Stages	Speed Zones (m)	Arbitrary (m)	Arbitrary CV (%)	Relative (m)	Relative CV (%)	Mean Difference (m)	95% CI of Mean Difference	ICC	F	*p*	*d*
Slow	Entire season	Z1	2118 ± 967	45.7	2237.6 ± 958.6	42.8	119.9	2025.4–2321	0.014	1.51	0.22	0.12
		Z2	837 ± 487	58.2	780.3 ± 460.1	59	56.3	751–864.4	0.004	1.37	0.243	0.12
		Z3	222 ± 160	72.1	206.1 ± 153.8	74.6	15.9	197.5–230.6	0.001	0.99	0.32	0.1
		Z4 *	111 ± 104	93.7	63.7 ± 71.3	111.9	47.4	68.2–106.2	0.037	28	**<0.001**	0.53
	Stage 1	Z1	1945 ± 533.4	27.4	2126.6 ± 577.5	27.2	181.9	1853.67–2221	0.081	3.51	0.063	0.33
		Z2	1097 ± 319.5	29.1	1018.1 ± 303.0	29.8	78.6	1002–1112.7	0	1.94	0.166	0.25
		Z3	297 ± 127.0	42.8	265.1 ± 120.8	45.6	32.2	259.2–303.2	0	2.06	0.154	0.26
		Z4 *	139 ± 89.1	64.1	67.2 ± 60.2	89.6	71.3	70.9–130.2	0.124	29.9	**<0.001**	0.94
	Stage 2	Z1	1881 ± 698	37.1	2011.4 ± 706.3	35.1	130.1	1816–2077	0.008	1.21	0.273	0.19
		Z2	825 ± 522	63.3	756.3 ± 489.7	64.7	68.4	707–874.3	0	0.64	0.425	0.14
		Z3	201 ± 157	78.1	189.0 ± 155.8	82.4	11.6	164.4–225.6	0.012	0.2	0.66	0.07
		Z4 *	106 ± 107	100.9	56.4 ± 64.1	113.7	50	61.8–101.2	0.024	11.5	**<0.001**	0.57
	Stage 3	Z1	2554.9 ± 1350	52.8	2602.0 ± 1339.5	51.5	47.15	2316–2837	0.007	0.04	0.845	0.04
		Z2	594.2 ± 459	77.2	573.3 ± 451.7	78.8	20.83	469–692	0.03	0.07	0.797	0.05
		Z3	172.3 ± 168	97.5	167.5 ± 165.3	98.7	4.78	140.5–199.2	0	0.03	0.874	0.03
		Z4	89.2 ± 110	123.3	68.3 ± 87.8	128.6	20.86	61.3–96.2	0	1.37	0.244	0.21
Fast	Entire season	Z1 *	1957 ± 814	41.6	2209.2 ± 801.7	36.3	252	2021–2202	0.031	16.5	**<0.001**	0.31
		Z2	1067 ± 513	48.1	1013.9 ± 518.2	51.1	53.4	882–1037	0.018	3.36	0.067	0.1
		Z3 *	314 ± 188	59.9	220.6 ± 149.3	67.7	93.5	228.5–270.6	0.022	46.4	**<0.001**	0.55
		Z4 *	162 ± 127	78.4	57.2 ± 60.4	105.6	104	94.5–125.7	0.041	213	**<0.001**	1.05
	Stage 1	Z1 *	1802 ± 461	25.6	2099.6 ± 532.7	25.4	297.9	1837–2063	0.078	25.4	**<0.001**	0.6
		Z2	1306 ± 365	27.9	1244.7 ± 361	29	53.4	978–1098.2	0.069	2.03	0.155	0.17
		Z3 *	378 ± 144	38.1	260.8 ± 125.1	48	116.8	246 - 288.9	0.101	57.6	**<0.001**	0.87
		Z4 *	188 ± 108	57.4	68.2 ± 54.8	80.4	120	107–149	0.1	143	**<0.001**	1.4
	Stage 2	Z1 *	1759 ± 656	37.3	2010.1 ± 666.2	33.1	251.5	1797.5–1969	0.004	9.51	**0.002**	0.38
		Z2	993 ± 495	49.8	935.9 ± 496.7	53.1	57	978–1098.2	0	2.03	0.155	0.12
		Z3 *	292 ± 181	62	200.8 ± 145.1	72.3	91.6	246–288.9	0	57.6	**<0.001**	0.56
		Z4 *	150 ± 135	90	47.1 ± 57.9	122.9	102.7	80.9–118.7	0.039	66.5	**<0.001**	0.99
	Stage 3	Z1	2385 ± 1113	46.7	2581.1 ± 1061.5	41.1	195.9	2243–2685	0.05	1.85	0.175	0.18
		Z2	868 ± 577	66.5	828.2 ± 600.1	72.5	39.9	768–928	0.002	0.25	0.617	0.07
		Z3 *	263 ± 220	83.7	195.7 ± 170.7	87.2	67.6	203–255.6	0	6.42	**0.012**	0.34
		Z4 *	144 ± 135	93.8	56.0 ± 67.5	120.5	87.8	83.4–117.4	0.016	37.4	**<0.001**	0.82

Note: F-statistic Linear Mixed Model; Z: Speed Zone; ICC: Intraclass correlation coefficient; CI: confidence interval; *p*: *p*-value; *d*: Cohen’s *d* effect size between Arbitrary and Relative demands; CV: coefficient of variation. * Statistical differences between relative and arbitrary approaches and shown in bold.

## Data Availability

The data are not available due to privacy or ethical restrictions.

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
