# Peer review of "Effect of Speed Threshold Approaches for Evaluation of External Load in Male Basketball Players"

_sensors, 2025, doi:10.3390/s25196085_

Round 1

Reviewer 1 Report

Comments and Suggestions for Authors

This study compared arbitrary vs. relative speed thresholds for monitoring external load in basketball, analyzing seasonal data from 12 male players to assess how methodological approaches influence workload interpretation across different locomotor profiles (fast/slow) and season stages. The authors concluded that relative speed zones reduced misclassification of high-intensity efforts (especially for fast players) and better individualized workload monitoring compared to arbitrary thresholds. Coaches should prioritize relative zones for precision but may retain arbitrary thresholds for group benchmarking.

The manuscript is well written and present interesting findings. However, some modifications are needed:

Introduction. Some recent studies could help strengthen the novelty of the present work:

Clemente et al. (2023) for threshold debate.

García et al. (2022) for training vs. match demands.

Participants. Add the calculation of the sample size.

Ensure consistency in formatting (e.g., km/h vs. km·h⁻¹)

Discuss future upgrades like AI analysis or integration with video.

Author Response

REV1. “Introduction. Some recent studies could help strengthen the novelty of the present work: Clemente et al. (2023) for threshold debate; García et al. (2022) for training vs. match demands.”
AUT. We have incorporated both suggested references in the Introduction to strengthen the rationale of the study. Clemente et al. (2023) were cited to emphasize the need for greater methodological rigor and replicability when applying individualized speed thresholds, recommending that population-specific and contextual factors should be considered rather than simply proposing alternatives to arbitrary cut-offs. Likewise, García et al. (2022) were included to highlight the relevance of high-speed running (>18 km·h⁻¹) as a differentiating variable, being the only one to surpass match demands during specific training sessions. These additions reinforce the novelty of our work by situating it within the current debate on speed threshold methodology and the training–competition continuum.

REV1. “Participants. Add the calculation of the sample size.”
AUT. An a priori sample size calculation was added to Methods – Participants, performed with G*Power (v.3.1.9.7). The calculation assumed a repeated-measures design with fixed (activity type, season stage, speed profile) and random (player identity) effects, a medium effect size (f = 0.25), α = 0.05, and power = 0.80, based on prior research in team-sport EL monitoring. Furthermore, in Strengths and limitations we now report a post hoc evaluation: power was ≥0.94 for large and very large effects observed in Zones 3 and 4 but fell below the conventional 0.80 threshold (~0.60) for moderate effects. While this suggests limited sensitivity for subtle differences, the robustness of results for high-intensity zones supports the adequacy of our sample for the main aims. We also note that future studies should increase sample size when aiming to detect smaller effect sizes or more complex interactions.

REV1. “Ensure consistency in formatting (e.g., km/h vs. km·h¹).
AUT. Velocity notation was standardized to km·h¹ throughout the manuscript, including text, tables, and figures.

REV1. “Discuss future upgrades like AI analysis or integration with video.”
AUT. In the Future lines of research section, we added a paragraph discussing the potential of artificial intelligence and video-based integration for external load monitoring. Specifically, AI can process large volumes of sensor data, identify the intensity and density of high-demand events, and generate more accurate load-control models, while video provides tactical context by linking locomotor efforts to specific game situations. Together, these technologies may support a more comprehensive and practical approach for coaching staff in training design, real-time monitoring, and decision-making.

Reviewer 2 Report

Comments and Suggestions for Authors

This manuscript presents a season-long observational study comparing arbitrary and relative speed thresholds for quantifying external load (EL) in male basketball players. Twelve semi-professional players were monitored using local positioning systems (LPS) across 13 matches and 45 training sessions. The study explored how different threshold methods affect the classification of players as fast or slow, interpretation of EL data, and seasonal performance tracking. Key findings suggest that arbitrary thresholds tend to overestimate high-speed activity—especially in faster players—and may misrepresent individual workload variations. In contrast, relative thresholds provide a more personalized view of player demands, supporting better monitoring and programming.

Major Comments

  1. The study is based on a small, homogeneous sample of 12 male players from one team.
  2. Suggestion: Acknowledge this limitation more prominently and recommend replication in larger, more diverse populations (e.g., elite players, youth, female athletes).
  3. The clustering method for defining "fast" and "slow" players is well-described, but lacks biomechanical or physiological validation.
  4. Suggestion: Include a brief justification or reference to validate this method (e.g., is 28.8 km/h a recognized benchmark for fast players?).
  5. While the manuscript touches on the idea of updating thresholds, it does not empirically test this.
  6. Suggestion: Either discuss a plan for dynamic updating in future work or simulate what effect updated thresholds might have had.
  7. The study focuses solely on external load.
  8. Suggestion: Consider discussing how the integration of internal load markers (e.g., heart rate, RPE) could enrich player monitoring.
  9. While Tables 3 and 4 are data-rich, they could benefit from improved readability.
  10. Suggestion: Add highlighting (e.g., bold significant values or effect sizes) and include a summary figure comparing EL differences across all zones.
  11. Some terms like "PlayerLoad" and the exact methodology for zone calibration could use clearer definitions.
  12. Suggestion: Include a glossary or figure that outlines key metrics and device specs for reproducibility.
  13. The manuscript is well-written, but occasional awkward phrases should be revised (e.g., “provided more precise and adaptable performance metrics” could be “provided more individualized metrics”).
  14. Ensure that all abbreviations are defined at first use (e.g., LPS, EL, MLM).
  15. Minor typos such as “exami-nation” and some formatting issues in the tables should be corrected.
  16. Some references are incomplete or incorrectly formatted (e.g., missing journal names). A final check is needed.
  17. Would updating relative speed zones monthly (instead of using static values) improve the sensitivity of load monitoring across the season?
  18. Have you explored whether positional roles (e.g., center vs. guard) influence EL readings differently under the two methods?
  19. Could your clustering approach be applied to other performance metrics (e.g., acceleration profiles)?
  20. How feasible is the implementation of relative zones in real-time coaching environments?
  21. What are your recommendations for practitioners with limited access to LPS systems?

Comments on the Quality of English Language

  • The English is technically accurate and professional, but the flow could benefit from light copyediting.
  • A few sentences are overly long or redundant. Shortening and simplifying these would improve clarity.
  • Figures, tables, and data presentation are well-organized but can be visually enhanced.

Comments on the Quality of English Language

Comments on the Quality of English Language

  • The English is technically accurate and professional, but the flow could benefit from light copyediting.
  • A few sentences are overly long or redundant. Shortening and simplifying these would improve clarity.
  • Figures, tables, and data presentation are well-organized but can be visually enhanced.

Author Response

REV2. “Small, homogeneous sample; recommend replication in larger, diverse populations.”
AUT. We now explicitly acknowledge this limitation in Strengths and limitations, noting that our sample comprised players from a single semi-professional male team. We recommend replication in larger and more diverse populations (different competitive levels, sexes, and age groups). Additionally, both a priori and post hoc power analyses were included to qualify the inference of our findings.

REV2. “Validate the clustering method/benchmark (e.g., is 28.8 km·h¹ recognized?).
AUT. We clarified that the classification into Fast and Slow profiles was data-driven, based on a k-means clustering procedure applied to peak velocity across the three season stages. This method minimized within-cluster variance and produced a centroid of 28.8 km·h⁻¹ for the Fast group, a value consistent with sprinting speeds previously reported in semi-professional and professional basketball. By relying on clustering rather than a fixed cut-off, we ensured a robust and context-specific classification that accounts for inter-individual variability in locomotor capacity.

REV2. “Updating thresholds not empirically tested; discuss a plan or simulate effects.”
AUT. In Future lines of research, we now recommend periodic recalibration of relative thresholds (e.g., monthly) as a potential strategy to enhance sensitivity. We also discuss how this could be combined with analyses of positional roles, clustering of other metrics (e.g., accelerations, decelerations, PlayerLoad), and the feasibility of real-time implementation with advances in wearable technologies. For practitioners without access to LPS, we suggest regulating the design of training tasks, particularly by limiting full-court, non-opposed drills, which are known to elicit higher physiological impact and volumes of high-intensity running.

REV2. “Only external load; consider integration of internal markers (HR, RPE).”
AUT. We added explicit mention in the Discussion (line 356) and in Future lines of research that combining external with internal load  would provide a more nuanced and ecologically valid interpretation of player demands.

REV2. “Tables 3 and 4: improve readability; add highlighting and a summary figure.”
AUT. Tables 3 and 4 were reformatted: significant differences are bolded, abbreviations clarified at first use, and new CV (%) columns were added. In addition, we created violin-plot summary figures to visually compare arbitrary vs. relative thresholds across speed zones and season stages.

REV2. “Terms like ‘PlayerLoad’ and zone calibration need clearer definitions; add a glossary/device specs.”
AUT. We developed a Glossary (Table 5, Supplementary Material) summarizing definitions of key concepts (e.g., external load, arbitrary vs. relative thresholds, speed profile, high-speed running, sprinting, seasonal stages) to ensure methodological transparency and reproducibility.

REV2. “Light copyediting; define abbreviations; fix typos and references.”
AUT. The full manuscript was carefully copyedited. Abbreviations are now defined at first use (e.g., EL, LMM, LPS, IMU). Typos and formatting errors in the tables were corrected, and references were completed and standardized according to journal style.

REV2. “Specific questions: monthly recalibration? positional roles? clustering for other metrics? real time feasibility? alternatives without LPS?”
AUT. In Future lines of research, we address all these points: (i) recalibration of thresholds on a monthly basis, (ii) possible influence of positional roles, (iii) extension of clustering to accelerations, decelerations, and PlayerLoad, (iv) real-time monitoring feasibility with emerging wearables, and (v) practical alternatives in the absence of LPS, emphasizing the regulation of full-court and non-opposed drills, which are known to induce greater physiological and high-intensity running demands.

Reviewer 3 Report

Comments and Suggestions for Authors

This manuscript compares the differences between fixed and relative speed thresholds in assessing external loads by tracking training and game data from 12 basketball players throughout the season. The results showed that fixed thresholds overestimated high-intensity running and underestimated low-intensity activities, especially for fast players. Relative thresholds, on the other hand, provide a more personalized description of the load, which is more suitable for long-term tracking of player status and fatigue management.
So, let's discuss some questions next. First, the sensors used in it were commercially purchased, which was not in line with the original intention of [Sensors]
In addition, the full text of this manuscript does not have a single figure description, which is incredible.
Therefore, I recommend refusing the publication of this manuscript

Author Response

REV3. “Commercial sensors not aligned with the journal’s scope; no figures; recommend rejection.”

AUT. We respectfully acknowledge the reviewer’s concern regarding the use of commercially available sensors. However, the selection of the Vector S7 (Catapult Sports) local positioning system (LPS) was intentional, given its widespread adoption in applied sports environments and its extensive validation for indoor team sports monitoring. Prior peer-reviewed studies have demonstrated its validity and reliability for quantifying distance, velocity, accelerations, and decelerations under competitive and training conditions (e.g., Hodder et al., 2020; Serpiello et al., 2018; Luteberget et al., 2018).

In this study, we further enhanced methodological rigor by ensuring that each player used the same device throughout the season to minimize inter-unit variability, and all monitoring sessions were conducted under standardized conditions on the same court. As noted by Torres-Ronda et al. (2022), the use of validated commercial systems can enhance both ecological validity and the applied relevance of findings in sports science. We therefore believe that employing this validated LPS technology is consistent with Sensors’ standards for methodological robustness and ensures that the findings hold strong applicability in real-world contexts.

Importantly, the methodological approach employed in this study also provides a distinct contribution to sensor-based athlete monitoring research. Unlike previous work limited to isolated matches, pre-season training, or short-term interventions, our study integrates longitudinal observations across a full competitive season, covering both training sessions and official matches. By combining repeated measures across seasonal stages with the classification of players into fast and slow speed profiles, we deliver a comprehensive perspective on how threshold selection influences external load quantification. This extended temporal scope, together with individualized profiling, enhances the practical relevance of the findings and supports a more context-sensitive interpretation of physical demands. In this way, the present study advances current methodologies and contributes original insights aligned with the technological and applied focus of Sensors.

Finally, we addressed the reviewer’s comment on the lack of figures by adding visual representations (e.g., violin plots of speed zones across groups and season stages) to complement the tables and enhance data interpretation.

We trust that these clarifications and additions align the manuscript more closely with the journal’s scope and address the reviewer’s concerns.

Reviewer 4 Report

Comments and Suggestions for Authors

This study aimed to i) identify differences in external load (EL) methodological approaches using arbitrary and relative running speed zones; (ii) examine the effect of the methodological approaches to identify fast and slow basketball players during competition and training; and (iii) determine the effect of the season stage on the methodological approaches. The issue is important because it improves the training process and athletic performance.

The article needs revision to improve its readability and scientific significance.

The title of the article should specify the sensors that are used to solve the scientific problem.

The introduction should be more structured and present a methodological approach to assessing performance indicators. Also, the methodology and devices for performance monitoring should be indicated.

In the Methodology section, you should first indicate the equipment and methods for measuring activity, and then indicate the parameters to be recorded.

In the statistics section, the comparisons made should be indicated.

The results section needs to be better structured.

The presentation of results in tables should be reviewed. Since the results have a large variability, the presentation as mean + SD is not correct.

In the Abstract, the obtained results must be specified by the specified parameters.

The abbreviation linear mixed model (MLM) is not accurate.

Overall conclusion: the manuscript is not ready for publication. Revision of the maintext and improvement of the methodology is necessary to increase the readability and relevance of the study.

Author Response

REV4. “Title should specify sensors used.”
AUT. We considered including the technology in the title (e.g., “Effect of speed threshold approaches for evaluation of external load in male basketball players using ultra-wideband local positioning system sensors”). However, after careful evaluation we decided to retain the original version of the title to avoid redundancy and ensure alignment with journal guidelines on conciseness of titles, while making sure that the Methods section clearly specifies the sensor technology employed.

REV4. “Introduction should be more structured; include methodological approach and devices.”
AUT. The Introduction was revised and strengthened, also in response to other reviewers, to improve structure and provide a clearer description of the methodological approach and the monitoring devices used, with additional references incorporated to contextualize the study.

REV4. “Methods: first equipment/methods of measurement, then recorded parameters; clarify statistical comparisons.”
AUT. Substantial adjustments were made in the Methods section to address this comment.

REV4. “Results section needs better structure; tables reconsider mean ± SD given high variability.”
AUT. Tables 3 and 4 were retained with mean ± SD for consistency, but a CV (%) column was added to quantify relative variability. Significant differences are now highlighted, and new summary figures were included to improve clarity and visual interpretation.

REV4. “Abstract: specify results by parameters.”
AUT. The Abstract was revised to report specific results, including quantitative differences between methods (~100 m in Z3–Z4; ~120–250 m in Z1; p < 0.001), training vs. match differences (+8.7% to +40.7% >18 km·h⁻¹), and the relative consistency of effects across seasonal stages.

REV4. “‘MLM’ abbreviation not accurate.”
AUT. The terminology was corrected throughout the manuscript, with “MLM” replaced by the accurate abbreviation “LMM” (Linear Mixed Model).

Round 2

Reviewer 2 Report

Comments and Suggestions for Authors

Thank you for addressing the comments.

Author Response

Comment 1: Thank you for addressing the comments.

Response 1: thanks for your positive feedback and suggestions that allowed to improve the manuscript.

Reviewer 3 Report

Comments and Suggestions for Authors

As I mentioned last time, all the sensors used in this paper were purchased. In my view, articles published in [Sensors] should contribute to the field of sensor technology. However, since all the sensors in this paper were bought, there is no innovation in the sensor aspect. I do not recommend publishing such papers in [Sensors].

Author Response

Comment 1: As I mentioned last time, all the sensors used in this paper were purchased. In my view, articles published in [Sensors] should contribute to the field of sensor technology. However, since all the sensors in this paper were bought, there is no innovation in the sensor aspect. I do not recommend publishing such papers in [Sensors].

Response 1: 

We appreciate the reviewer’s continued engagement. However, we respectfully disagree with the assertion that purchased sensors preclude scientific innovation or contribution to the field of sensor technology, particularly in the context of the Special Issue entitled: “Development of sensors and technologies for exercise monitoring; validation of signals and metrics recorded with wearable sensors during exercise; applications of sensor technologies in sports and exercise.”

This Special Issue, as introduced by the Guest Editors, explicitly encourages studies that validate or apply commercial sensors in the real-world settings of sport and exercise, to address the growing concern that technological tools are often used without rigorous testing, potentially leading to misleading metrics and suboptimal decision-making in training and health management.

Our manuscript directly contributes to this aim by implementing and critically analyzing commercial wearable sensor data over a full competitive basketball season using advanced methodological frameworks. This includes:

  • Longitudinal tracking of players' peak speed profiles,

  • Comparison of arbitrary versus relative thresholding approaches,

  • Training versus match-based load comparisons,

  • Interpretation of high-speed demands across seasonal stages.

This type of translational research is essential to bridge the gap between sensor development and field-based implementation, thus aligning with the journal’s and Special Issue’s objectives.

Moreover, the journal Sensors has a strong record of publishing papers that used purchased sensor systems to deliver novel contributions in sport science, performance assessment, and load monitoring. Below, we provide several examples from both basketball and football, all of which used commercial sensors but offered value through validation, methodology, or application:

Football Studies:

  1. Piłka, T., Grzelak, B., Sadurska, A., Górecki, T., & Dyczkowski, K. (2023). Predicting injuries in football based on data collected from GPS-based wearable sensors. Sensors, 23(3), 1227. https://doi.org/10.3390/s23031227

  2. Calderón-Pellegrino, G., Gallardo, L., Garcia-Unanue, J., et al. (2022). Physical demands during the game and compensatory training session (MD+1) in elite football players using global positioning system device. Sensors, 22(10), 3872. https://doi.org/10.3390/s22103872

  3. Garcia-Unanue, J., Hernandez-Martin, A., Viejo-Romero, D., et al. (2024). The impact of a congested match schedule (Due to the COVID-19 lockdown) on Creatine Kinase (CK) in elite football players using GPS tracking technology. Sensors, 24(21), 6917. https://doi.org/10.3390/s24216917

  4. Hernandez-Martin, A., Sanchez-Sanchez, J., Felipe, J. L., et al. (2020). Physical demands of U10 players in a 7-a-side soccer tournament depending on the playing position and level of opponents using GPS. Sensors, 20(23), 6968. https://doi.org/10.3390/s20236968

Basketball Studies:

  1. Espasa-Labrador, J., Martínez-Rubio, C., Oliva-Lozano, J. M., et al. (2024). Relationship between Physical Demands and Player Performance in Professional Female Basketball Players Using Inertial Movement Units. Sensors, 24(19), 6365. https://doi.org/10.3390/s24196365

  2. Espasa-Labrador, J., Fort-Vanmeerhaeghe, A., Montalvo, A. M., et al. (2023). Monitoring Internal Load in Women’s Basketball via Subjective and Device-Based Methods: A Systematic Review. Sensors, 23(9), 4447. https://doi.org/10.3390/s23094447

  3. Flórez-Gil, E., Vaquera, A., Conte, D., & Rodríguez-Fernández, A. (2025). Quantifying the Effects of Detraining on Female Basketball Players Using Physical Fitness Assessment Sensors. Sensors, 25(7), 1967. https://doi.org/10.3390/s25071967

We believe these examples reflect a consistent editorial precedent and support the relevance of our study to both the journal and the Special Issue. Thus, while we acknowledge that our sensors were commercially sourced, the scientific contribution of our manuscript lies in methodological innovation, application context, and real-world translational value.

Reviewer 4 Report

Comments and Suggestions for Authors

the manuscript has been sufficiently improved to warrant publication in Sensors

Author Response

Comment 1: the manuscript has been sufficiently improved to warrant publication in Sensors

Response 1: thanks for your positive feedback and fruitful comments that allowed to improve the manuscript.